

# A geomorphological slope unit dataset for the eastern edge of Tibetan Plateau

Xiangyi Zheng[1,2], Ying Wang[1,2], Qigen Lin[3], Jing Qi[1,2], Yuxin Li[1,2], Mengxia Zhao[1,2], Xinren Zhang[1,2], Xia Wang[1,2], Yu Chen[1,2]

5  [1]Key Laboratory of Environmental Change and Natural Disaster of Ministry of Education, Faculty of Geographical Science, Beijing Normal University, Beijing, 100875, China
[2]Academy of Disaster Reduction and Emergency Management, Ministry of Emergency Management and Ministry of Education, Beijing Normal University, Beijing, 100875, China
[3]Institute for Disaster Risk Management, School of Geographical Sciences, Nanjing University of Information Science & 10  Technology, Nanjing, 210044, China

*Correspondence to*: Ying Wang (wy@bnu.edu.cn)

**Abstract.** Geomorphological slope units represent the polygonal units on the digital terrain that are jointly segmented by valley and ridgelines. In contrast to the grid cells commonly used in traditional spatial analyses, slope units have an explicit geomorphological and environmental implication and capture the topographic characteristics of different units comparatively 15  faithfully, which is being increasingly extensively used in the investigation of natural hazards, ecological processes and environmental impacts. However, delineating slope units on a widespread regional scale remains challenging, especially in areas such as the eastern edge of Tibetan Plateau, which is characterized by considerable spatial heterogeneity of topography and fragile ecological environments. To enable more researchers to focus more conveniently on the subject matter to be addressed itself, rather than being caught up in the slope unit delineation. The present study delineates and produces a dataset 20  of high-precision geomorphological slope units for the eastern edge of the Tibetan Plateau based on the publicly available high-resolution DEM data. A total of 446,497 slope units were derived, representing an area of 350,000 square kilometres. To facilitate the application of this geomorphological slope unit dataset by researchers, we use it for landslide susceptibility assessment and perform an insightful evaluation and comparison with the results of the traditional mapping units. The dataset of this geomorphological slope unit demonstrates good performance in terms of overall scale, sample scale and unit 25  scale. It is available in Figshare at https://doi.org/10.6084/m9.figshare.24457144.v1(Zheng et al.,2023) and could be used as fundamental data for the investigation of disasters, environment and ecology in the eastern edge of the Tibetan Plateau.

## 1 Introduction

Slope units are the area delineated by drainage lines and watershed lines, which are the basic topographic units of natural geological hazards (Wang et al., 2017). Slope units are usually used in the hazard evolution, such as landslide susceptibility, 30  debris flow risk and flood risk, etc (Deng et al., 2022; Huang et al.,2018). Due to the intense tectonic activity and complex



topography of the eastern edge of Tibetan Plateau, the deformation and failure of steep slopes are prone to slide (Dai et al., 2019, Zhao et al., 2021; Zhan et al., 2018).

In the field of geologic hazard research, the mapping units regarded as the sampling units are diverse (Rotigliano et al. 2012), such as grid units (Du et al.,2019), slope units (Hua et al.,2021) and unique-condition units (Domenico et al.,2010). Grid units are the regular square cells with specified size (Cama et al. 2016). They can be stored in the matrix format for convenient calculation, so grid units are the most frequently used mapping units (Ba et al., 2018; Paola et al.,2018). But the grid units are not associated closely with the geological environments (Guzzetti et al. 1999; Erener, A.&Düzgün, H.S.B., 2012). Unique-condition units are delineated by overlaying various geologic hazard impact factor classification maps (Chiessi et al. 2016). The scale of units depends on the quantity of impact factors, and their total number depends on the classification standard of impact factors (Ba et al., 2018). However, the shortcoming of unique-condition units is that the classification standard of impact factors is subjective (Carrara and Guzzetti 1995). Therefore, slope unit is a map unit with both objectivity and closely association with geological environment, which is suitable for geological disaster research.

As the increased frequency of using slope units, some studies explored the efficient and objective extraction method of slope units. At first, the slope units were delineated manually from topographic maps, which is subjective, time-consuming and limits their scale of application (Carrara, 1988; Alvioli et al., 2016). With the development of GIS, some studies used the hydrology tools of ArcGIS to extract the ridge and valley lines to generate the slope units (Xie et al., 2004; Erener and Düzgün 2012). The automatic delineation of slope units with r.slopeunits software was presented to product the reliable and reproducible slope zonation (Alvioli et al., 2016). With the morphological image analysis, the new method of extracting homogeneous slope units was revealed used logical algorithms (Wang et al., 2019). Existing studies have proposed various slope unit extraction methods, but the source and resolution of original data used are different (Tian et al.,2019). Moreover, the extracted slope units only cover one watershed, making it difficult to form a dataset for other studies. The eastern edge of Tibet Plateau is vast and prone to geological disasters, which need to carry out geological disaster studies (Du et al.,2017). Therefore, a comprehensive slope unit dataset with the same extraction standard is needed as the basic data for studying geological hazards (Guzzetti et al. 2005).

Existing research have assessed and mapped landslide susceptibility based on slope unit in the Tibetan Plateau and surrounding mountainous areas (Sun et al.,2020; Wang et al.,2017). But, when extracting slope units from high spatial resolution DEM, the study area has only been chosen as a small area like a catchment or a watershed (Li & Lan et al.,2020; Wang et al.,2020). For large-scale study area, slope units mostly extracted from DEM with a small scale is less precise (Tian, S. & Kong, J.,2013; Gregory C. O., & John C. D., 2003; Liu et al., 2021). Accordingly, few high-precision slope unit datasets could cover large area to assess geological disaster (Chung, C.-C. & Li, Z.-Y, 2022; Qiu et al., 2005). The selection of accurate slope units is crucial to the assessment of geological disaster (R. Schlögel et al., 2018; Guzzetti et al. 2005). Aimed to convenient for geological hazard research, this study constructs a slope unit dataset within a large area to overcome the limitations of their application in small areas. And we took landslide susceptibility assessment as an example to illustrate the advantages of our slope unit dataset.





In this study, through 12.5m high-resolution DEM data and hydrology tools of ArcGIS, the high-precision slope units of the eastern edge Tibetan Plateau were extracted. And the error part of slope units was manually modified which were delineated from the water surface. Based on the 12.5m grid unit and the high-precision slope unit in this dataset, the landslide susceptibility of the eastern edge Tibetan Plateau was assessed with the logistic model. Then, the landslide susceptibility results of the two were compared at the overall scale, sample scale, and unit scale. It is expected that this dataset will

contribute to future geological hazard research in the hazard-prone area of the eastern edge Tibetan Plateau.

## 2 Study area

The study area extends for 350 thousand km$^2$ along the eastern edge of the Tibetan Plateau, which is the administrative region of Sichuan Province in China. The western part of Hengduan Mountain extends along the study area from north to south and is located in the Mediterranean–Himalayan volcanic seismic belt.

The eastern edge of the Tibetan Plateau is the transition zone between the Panic Rim and the ancient Mediterranean, which is one of the most complex geological structures in China (Pan, 1989). Squeezed by the collision of the Eurasian, Indian and Pacific plates, complex deep fault zones distributed from north and south are the result of intensive neo-tectonic movement and seismic activity in the study area (Xu et al., 2019). The eastern edge of the Tibetan Plateau is characterized by a wide variety of formations and lithologies, where Quaternary sediments are extensively developed, with outcrops of fluvial–

lacustrine and lake–marsh sediments, mainly composed of sandy soil and sandy gravel (Yin et al., 2001; Zhang et al., 2006). The vertical and horizontal distributions of rivers and dense tributaries are the reasons for intense river erosion and deep canyon landforms, with 81.5% of the sector having a relative height difference of more than 1000 m (Li, 1989; Bian et al., 2018). Accordingly, deeply cut fault zones, abundant surface deposits and intense river erosion are the main causes of landslides.

The eastern edge of the Tibetan Plateau spans subtropical and plateau temperate areas, with dry winters, wet summers and obvious wet–dry seasons. However, they have changeable local climates affected by complex terrain. This study region is mainly influenced by the Northern Hemisphere westerly circulation and monsoons, in which the monsoon is the main air mass for precipitation (Wei et al., 2018).

In response to the litho-structural and hydroclimatic setting of this region, landslides are widespread (Fig. 1). They are

90 mostly fast-moving slide-type and flow-type movements, and rapid-moving landslides are also abundant. The widespread occurrence of landslides in this area is near roads, rivers and residential regions, which cause casualties and economic losses. These characteristics have a direct impact on the development and economy of Sichuan Province's mountainous areas.





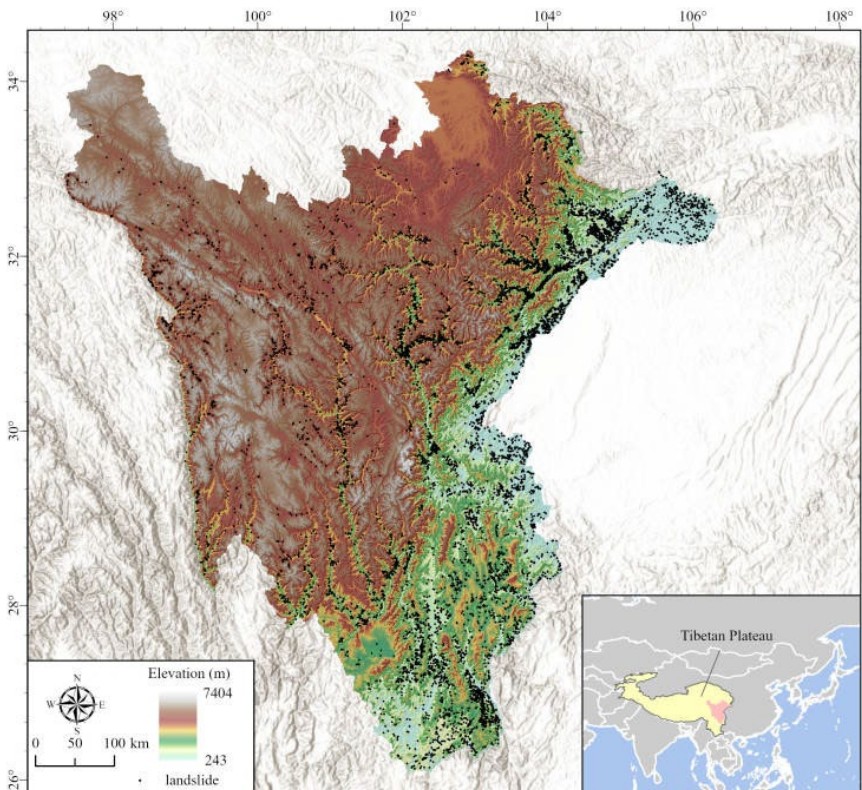

**Figure 1: Location map of the study area and landslide inventory. The hillshade map comes from Esri USGS.**

## 3 Methodology

### 3.1 Data source and landslide inventory

The main DEM data in this study are a 12.5 m high-resolution DEM collected by phased array L-band synthetic aperture radar (PALSAR) from the Advanced Land Observing Satellite (ALOS); river data from the Opening Street Map; fault data from the 2011 Atlas of Natural Disaster Risk in China; precipitation, temperature, NDVI (normalized difference vegetation index), GDP (gross domestic product) and land use grid data provided by the Resource and Environment Science and Data Centre of the Chinese Academy of Science and lithology vector data from ISRIC-World Soil Information and FAO. Table 1 shows a detailed list of the resolutions and resources of the datasets used in this research. All datasets were resampled to a resolution of 12.5 m before inputting the calculation. All the above datasets are available free at the URLs listed in Table 1.

The landslide inventory includes an area of 350 thousand km² and 9579 landslides corresponding to an average density of approximately 0.027 landslides per square kilometre. The landslide dataset used in this study was provided by the project "The 14th Five-Year Plan for National Geological Disaster Prevention and Control" dating until 2020, mapped at a scale of



1:2,000,000 (Fig. 1). The locations of the landslides in the inventory are shown in Fig. 1, and they are generally distributed below 4000 meters above sea level and along the river valley. The landslide distribution is denser in the east.

**Table 1.** Details of the data of factors influencing landslides

| Factors | Resolution | Source | Reference |
|---|---|---|---|
| Elevation | 12.5 m | https://asf.alaska.edu | |
| Aspect | 12.5 m | — | |
| Slope | 12.5 m | — | |
| Relief | 12.5 m | — | |
| Curvature | 12.5 m | — | Yilmaz (2009); |
| SPI | 12.5 m | — | Pourghasemi and |
| TWI | 12.5 m | — | Rahmati (2018); Basu |
| Distance to river | 600 m | https://www.openstreetmap.org/#map=7/23.611/120.768 | and Pal (2018); Jones et al. (2021); Eker and |
| Distance to fault | 600 m | 2011 Atlas of Natural Disaster Risk in China (Shi P., 2011) | Aydin (2014); Ballabio and Sterlacchini (2012); Sujatha and |
| Snow depth | 500 m | https://www.doi.org/10.12072/ncdc.I-SNOW.db0011.2021. | Rajamanickam (2011); Lee et al. (2017); Akgun |
| Precipitation | 1 km | http://www.resdc.cn/10.12078/2022090901 | (2012); Youssef (2015); |
| Temperature | 1 km | http://www.resdc.cn/10.12078/2022090902 | Talaei (2014); Haque et |
| NDVI | 1 km | http://www.resdc.cn/10.12078/2018060601 | al. (2019) |
| GDP | 1 km | http://www.resdc.cn/10.12078/2017121102 | |
| Land use | 1 km | http://www.resdc.cn/10.12078/2018070201 | |
| Lithology | 13 m | http://www.isric.org/isric/Webdocs/Docs/ISRIC_Report_2008_06.pdf | |

## 3.2 Conditioning factors

With the geographical environment of the study area, 16 factors influencing landslides were selected, and they were mainly divided into two categories: geographical environmental factors and human activity factors.



### 3.2.1 Geographical environmental factors

DEMs are the most commonly used factors in landslide susceptibility analysis, which is indicative of topography and landform morphology. The slope is the angle formed by any earth surface with level, and it is a significant factor impacting the shear stress of rock and soil on slopes. The slope aspects range from 0° to 360°, while the aspect of the flat area is set to -1. The slope in each direction has different rainfall conditions and solar radiation intensities. Curvature, defined as the slope of a slope, affects the forces of slope material and the movement of runoff. Topographic relief represents the altitude difference between the highest and lowest points in a unit area and is used to quantitatively describe the topographic features, which have a great correlation with landslides. The stream power index measures the spatial intensity of runoff, indicating the effect of the water flow. The topographic wetness index quantifies the topographic control of hydrological processes and characterizes local soil moisture conditions. The stream network can affect landslide susceptibility, and the distance to the river is the Euclidean distance to the river calculated in ArcGIS. The active fault zone triggers the landslide, and the distance to the fault is calculated in the same way with ArcGIS. Vegetation coverage is represented by the NDVI (normalized difference vegetation index), which is calculated with the reflectivity of the near-infrared and red bands. The mean annual temperature data are represented by the temperature index. Snow thickness is represented by the annual average snow thickness data. Lithology reflects the characteristics of rocks, and rocks with unstable structures are more prone to landslides. Rainfall is usually the factor inducing landslides, so the average annual rainfall is set as the susceptibility factor.

### 3.2.2 Human activity factors

GDP has a strong correlation with human activities, so this study selects GDP as the landslide factor indicating the intensity of human activities. In regions with a high degree of land use, the original geological environment is destroyed, resulting in a more unstable slope. Therefore, land use is considered the factor influencing landslide susceptibility. Cultivated land with more active human activities and damaged bare land are more prone to landslides.

### 3.3 Logistic regression model

Logistic regression is an extension of multiple regression and is suitable for cases where the dependent variable is not a quantitative variable or a continuous variable (George and Mallery 2000). Therefore, the dependent variable can be regarded as a binary variable (such as the existence or nonexistence of landslide hazard points). Moreover, the advantage of the regression is that the independent variable can be either continuous or discrete (dummy variable) in the statistical analysis and allows us to build a nonlinear model (Mertler & Vannatta 2002). Since 2000, research on landslide susceptibility analysis with logistic regression has increased year by year, accounting for approximately 18.5% of the literature database, far exceeding the application frequency of other models in landslide susceptibility analysis (Paola et al. 2018). Therefore, this paper studies the relationship between landslide observations and factors of influence with a binary logistic regression model. Logistic regression generally fits the dependent variable using an equation of the following Eq. (1):



$$P = \frac{\exp(\beta_0 + \beta_1 x_1 + \beta_2 x_2 + \cdots + \beta_i x_i)}{1 + \exp(\beta_0 + \beta_1 x_1 + \beta_2 x_2 + \cdots + \beta_i x_i)} \qquad (1)$$

$P$ is the probability of the dependent variable occurring, where the value is [0,1], indicating the possible probability of the presence of the landslide. $x_i$ are independent variables related to influential factors (i=1,2,3......); $\beta_0$ is constant, which is the intercept of the partial regression function. $\beta_i$ are partial regression coefficients (i=1,2,3......), which reflect the degree of influence of the independent variable factor $x_i$ on P, and their positive and negative values illustrate the effect and degree of the independent variable on the dependent variable. When $\beta_i$ is greater than 0, the independent variable is positively

correlated with the dependent variable. Otherwise, $\beta_i$ less than 0 represents a negative correlation between the independent and dependent variables.

Since the number of units with landslide hazards is 9579 and that without landslide hazards is 441159, landslide events account for 2.17% of non-landslide events, which can be considered rare events. Within various studies, the landslide pixels in the model training dataset are expanded to ratios of 1:1, 1:2, or 1:5 to the non-landslide pixels for susceptibility analysis

(Basu, T., & Pal, S, 2017). According to the geographical characteristics of the study area, this paper chooses the ratio of landslide unit expansion to non-landslide units as 1:1.

In this study, due to the large span of the study area, the latitudinal span is approximately 8°, and the longitudinal span is approximately 9°. Large differences exist in the geographical environment, so the overall modelling accuracy is low. Therefore, according to the main types of internal and external forces of the landform (the third-level zoning of the

landform), landslide susceptibility modelling is carried out based on dividing the study area into three parts, namely, regions A, B, and C. The landforms in A are mainly extremely high mountains and high valleys, those in B are mainly alpine valleys, and those in C are mainly high mountains and plateaus (Cheng et al., 2019; Li et al., 2013).

# 4 Generation of slope units

In landslide susceptibility analysis, commonly used basic assessment units include grid units, slope units, watershed units,

and unique condition units (administrative units). In existing studies, approximately 86.4% use grid units as mapping units (Paola et al.,2018). The frequency of use of other evaluation units is much lower than that of grid units. The frequency of use of slope units and unique condition units is less than 5%. The high usage rate of grid units is because most of the geographical factor data are stored in raster form, so it is easy to process at various resolutions and scales. The landslide susceptibility based on the grid units is inconsistent with the actual landslide distribution pattern, limiting by the geographic

irrelevance. When evaluating landslide susceptibility, using slope units is more consistent with the geographical distribution of landslides, and the susceptibility results are also more characteristic of terrain trends than grid units.

This dataset uses DEM data for water system extraction and sub-basin segmentation. The sub-basin boundaries extracted by elevation are used as ridge lines, and the sub-basin boundaries extracted by inverted elevation are used as valley lines. Valley line and ridge line layers are superimposed to obtain slope units. The above steps can be implemented in the Spatial Analyst

Tools-Hydrology toolbox of ArcGIS. The specific steps are as follows: ① First, the depression is filled by elevation data to allow all water flows within the area to flow out from the boundary. The flow direction is calculated based on the elevation after filling, and then the upstream catchment area is calculated based on the flow direction as the flow rate. The river network is treated as formed after the flow reaches a certain threshold, and the threshold of the number of upstream water collection units is set to extract the river network. The river network is decomposed into branches according to nodes, which

serve as local confluence depressions. The sub-basin is calculated using the flow rate as a reference and the catchment depression as the water collection point. The sub-basin raster is vectorized to obtain the sub-basin boundary and obtain the ridge line. ② After inverting the elevation, step ① is repeated to obtain the boundary line of its sub-basin and obtain the valley line. Finally, the two-vector data are combined to obtain the slope unit data. This method has errors in identifying large areas of water surfaces. Therefore, this dataset is manually corrected and checked based on the shaded terrain map for

all water surfaces in the slope unit.

The slope units in this dataset cover a total of 350,000 square kilometres, with a total of 446,497 units. The number of small units in each area range is shown in Fig. 2(a). The number of slope units in level III (0.4-0.6 km²) is the largest, accounting for approximately 24.41%. Figure 2(b) shows the slope unit at the local scale, with different areas classified by different colours. This dataset is intended to promote slope units in landslide-prone mountainous areas in eastern Tibetan Plateau so

that more research on landslide susceptibility can be conducted here.

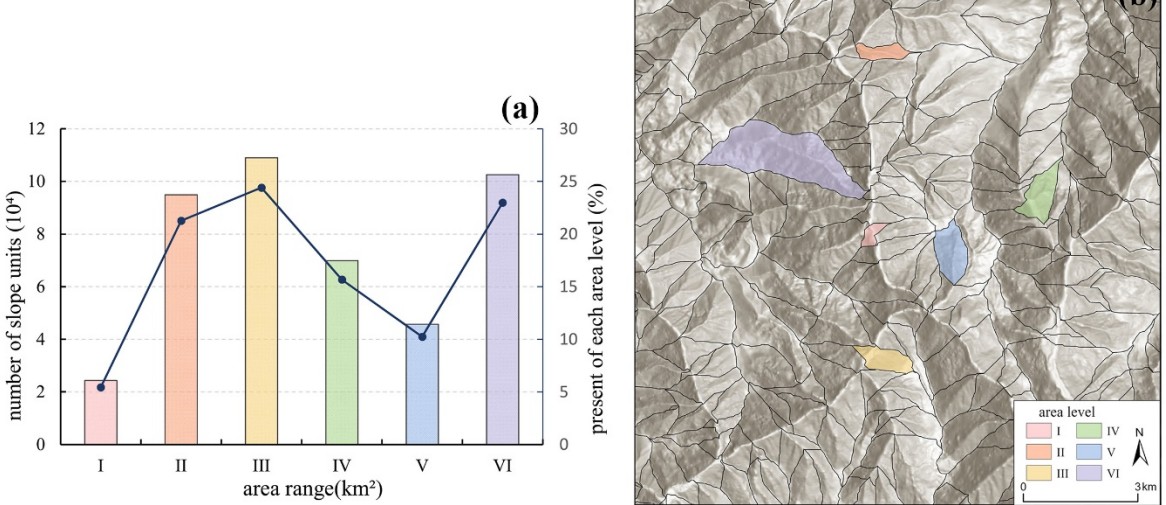

**Figure 2: (a) Bar chart of the number of slope units in each level of area (I: 0-0.2 km², II: 0.2-0.4 km², III: 0.4-0.6 km²; IV: 0.6-0.8 km²; V: 0.8-1 km²; VI: >1 km²); (b) local slope unit map and units under each level of area. The hillshade map comes from Esri USGS.**



## 5 Difference between the slope and grid units

**5.1 Comparison based on the whole-region scale**

The landslide susceptibility distribution based on the slope units in the high mountainous area of Sichuan is shown in Fig.
3(a). Units with an occurrence probability greater than 90% are classified as high-susceptibility units, and there are
approximately 13,059 units, accounting for approximately 2.90% of the total number of units. Units with an occurrence
probability greater than 80% and less than 90% are medium–high-susceptibility units, and there are approximately 21,029
units, accounting for approximately 4.67% of the whole. Those with occurrence probabilities greater than 70% and less than
80% are medium–high-susceptibility units, and there are approximately 22,191 units, accounting for approximately 4.92% of
the whole. Those with occurrence probabilities greater than 50% and less than 70% are approximately 45,490 units of
medium and high susceptibilities, accounting for approximately 10.09% of the whole. Units with an occurrence probability
of less than 50% are medium- and high-susceptibility units, and there are approximately 348,969 units, accounting for
approximately 77.42% of the whole.

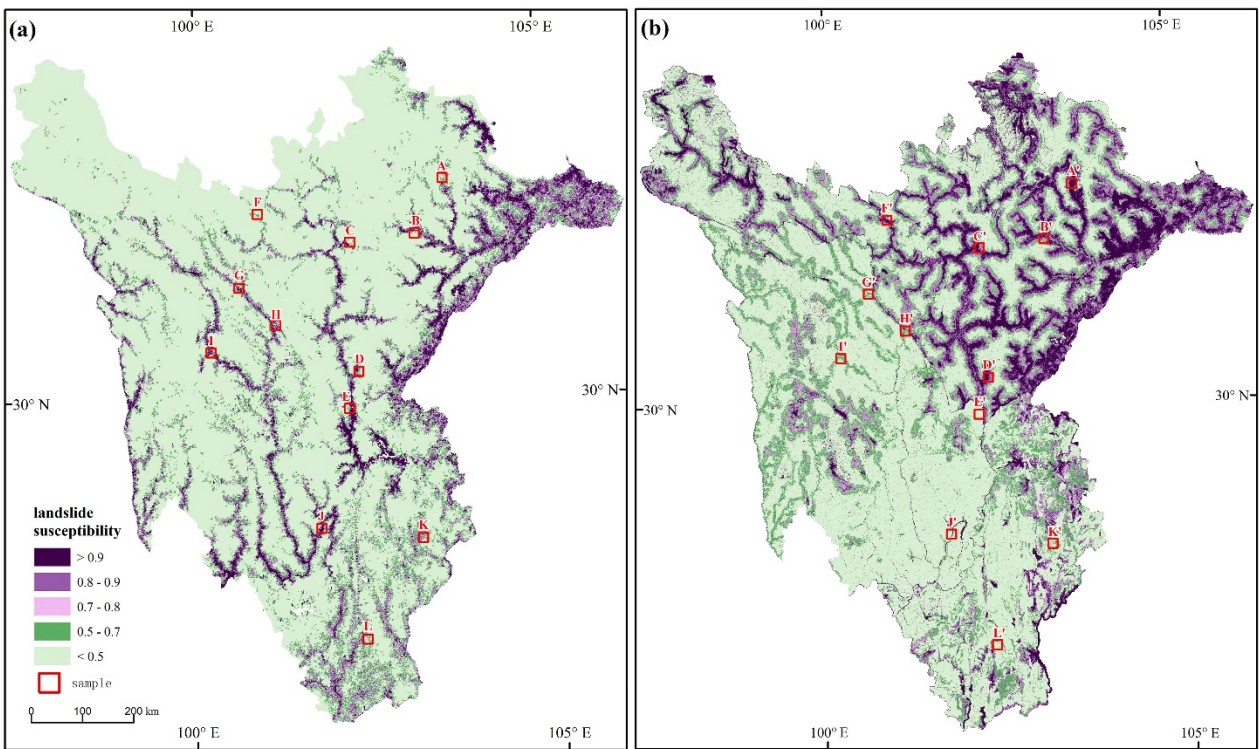

**Figure 3: (a) Landslide susceptibility map based on slope units, (b) landslide susceptibility map based on grid units. A-L are 12
randomly selected enlarged samples.**

Overall, slope units with high landslide susceptibilities are distributed along the river valley, which is very similar to the
original trend of landslide point distribution in Fig. 1. This distribution pattern is very significant in the Western Sichuan





Plateau. Due to side bank erosion of rivers in river valleys, the slope toes on both sides of the river banks erode, leading to slope instability and landslides; for terrain reasons, the water vapor content in the air in the valley is high, and cold air masses flow into the valley bottom to force warm air to rise, easily causing rainfall. Therefore, geological disasters such as
landslides are common in river valleys. Units with medium to high landslide susceptibilities are clustered along the eastern edge of the Tibetan Plateau, with undulating mountain ranges and in the transition zone between the Western Sichuan Plateau and the Sichuan Basin. Because the geological tectonic activities along the eastern edge of the Tibetan Plateau and transitional terrain areas are very active and the slope stability is poor, this area is prone to landslides and other geological disasters.

The landslide susceptibility distribution based on the grid units (spatial resolution 12.5 m) in the high mountainous area of Sichuan is obtained, as shown in Fig. 3(b). Units with an occurrence probability greater than 90% are high-susceptibility units. There are approximately $1.77 \times 10^8$ such units, accounting for approximately 8.80% of the total number of units. Units with occurrence probabilities greater than 80% and less than 90% are medium–high-susceptibility units. There are approximately $1.85 \times 10^8$ units, accounting for approximately 9.17% of the whole; units with occurrence probabilities greater
than 70% and less than 80% are medium–high-susceptibility units, and there are approximately $1.66 \times 10^8$ such units, accounting for approximately 8.22% of the whole; those with occurrence probabilities greater than 50% and less than 70% are medium–high-susceptibility units, and there are approximately $2.80 \times 10^8$ such units, accounting for approximately 13.92% of the total; those with occurrence probabilities less than 50% are medium–high-susceptibility units. There are approximately $1.21 \times 10^9$ such units, accounting for approximately 59.89% of the total. Grid units with high landslide susceptibilities are
distributed along the river network in the north-eastern part of the study area and along the river valley in the south-eastern part of the study area. The trend of grid units with high landslide susceptibility distributed in the western part of the study area is not significant. The overall landslide susceptibility distribution pattern based on grid units is quite different from the original landslide point distribution trend in Fig. 1.

Twelve quadrats with a side length of 12.5 km were randomly selected within the study area. Figure 4 shows the
amplification of landslide susceptibility results for 10 groups of quadrats A-L (except quadrats D and I) based on slope and grid units. The amplification of the results of quadrats D and I is shown in the comparison of quadrat scales in the next section. Figure 4 shows that there are overestimated discriminations in quadrats A', B', C' and F' based on grid units compared with quadrats A', B', C' and F based on slope units; most of the overrated units have high landslide susceptibilities, and the locations of the high-susceptibility units are basically river valleys. The high landslide susceptibility in the valley is
not conducive to the development of landslide control measures and may cause excessive prevention in the valley area. The other grid quadrats (E', G', H', J', K', and L') are underestimated compared with the slope units and can identify only the high landslide susceptibility at the river, but it is difficult to identify the middle- and high-grade landslide susceptibilities. Most of the grid units are identified as low, middle-low grade landslide susceptibilities. Compared with the 10 quadrates, the grid unit tends to be overly high or low for assessing landslide susceptibility, so the performance of slope units is better than that
of grid units in landslide susceptibility assessment.



**Figure 4: Magnification of the contrast in quadrat A-L (except D and I) based on slope and grid units.**



The above results show that when the probability of landslides is greater than 50%, for the proportion of units at each
landslide-prone level, grid units are larger than slope units. Comparing Fig. 3 (a) and (b), the grid units in the north-eastern
part of the study area have a greater impact on landslides. There is overestimation in susceptibility discrimination.
Comparing the landslide susceptibility map results based on the two units, the trend of the distribution of high landslide
susceptibilities based on the slope units is more consistent with the measured landslide distribution pattern than that of the
high susceptibilities based on the grid units and is consistent with the measured landslide point distribution. In contrast, there
is an underestimated assessment of landslide susceptibility in the southwestern part of the study area in the grid unit.
According to Table 2, based on two different evaluation units, among the zoning modelling accuracies of areas A, B, and C
in the study area, the accuracy of the slope unit is higher than that of the grid unit, which is approximately 0.3% to 6.0%
higher. Therefore, when analysing landslide susceptibility in large-scale areas, based on slope unit modelling, the landslide
susceptibility distribution can better reflect the trend. The trend is more in line with the natural pattern of measured landslide
distribution and the significant pattern of distribution along the river valley. In addition to not overjudging landslide
susceptibility, the modelling accuracy based on slope units is higher in large-scale areas.

**Table 2.** Table of partition modelling precision

| Region | Observed | Slope unit | | | | Grid unit | | | |
| | | Predicted | | Precision | Accuracy | Predicted | | Precision | Accuracy |
| | | Landslide | No landslide | | | Landslide | No landslide | | |
|---|---|---|---|---|---|---|---|---|---|
| A | Landslide | 181740 | 22516 | 89.0 | | 164684 | 39572 | 80.6 | |
| | No Landslide | 40793 | 161340 | 79.8 | 84.4 | 48133 | 153997 | 76.2 | 78.4 |
| B | Landslide | 125712 | 24084 | 83.9 | | 123228 | 26568 | 82.3 | |
| | No Landslide | 27370 | 121129 | 81.6 | 82.7 | 39084 | 109415 | 73.7 | 78.0 |
| C | Landslide | 74920 | 20840 | 78.2 | | 75640 | 20120 | 79.0 | |
| | No Landslide | 27095 | 65315 | 70.7 | 74.5 | 28325 | 64085 | 69.3 | 74.2 |




## 5.2 Comparison based on the sample-area scale

Figure 5 (D) and (D') show the overestimation of landslide susceptibility by grid units compared with slope units, which are typical of 12 quadrats. By comparison, it is found that the grid unit is too high to identify the landslide susceptibility;
although the original landslide points are in the range of high landslide susceptibility, it does not identify the low landslide susceptibility region. However, the classification of slope units for each grade of landslide susceptibility is relatively accurate, and there is no discrimination of overly high or low. The range of landslide susceptibility results based on slope units all coincide with the original landslide points, but there are some high landslide susceptibility units that do not have actual landslide points.

In this paper, Fig. 5(a), (b) and (c) are used to indicate that the slope unit is better than the grid unit in determining landslide susceptibility, and the slope unit can predict the possible probability of landslides more accurately and has excellent performance in non-landslide areas. Based on the results of high landslide susceptibility based on the slope unit, satellite remote sensing images in Google Maps were used to visually interpret the possible landslide locations within the unit, as shown in Fig. 5(a) and (b). A total of 81.82% of slope units with high landslide susceptibilities and 76.92% of slope units
with medium and high landslide susceptibilities have possible landslide sites. Although some of the slope units with high and medium landslide susceptibilities do not have actual landslide sites, the possible landslide sites within the unit range can be identified by high-resolution satellite remote sensing images. Therefore, the slope unit has a better performance in identifying high- and medium–high-landslide susceptibilities. However, Fig. 5(c) shows that there is much overlap between the medium–high landslide susceptibility range of the grid unit and the low landslide susceptibility range of the slope unit.
According to the actual observed landslide points, there is no overlap between the slope unit and the low landslide susceptibility range of the slope unit.

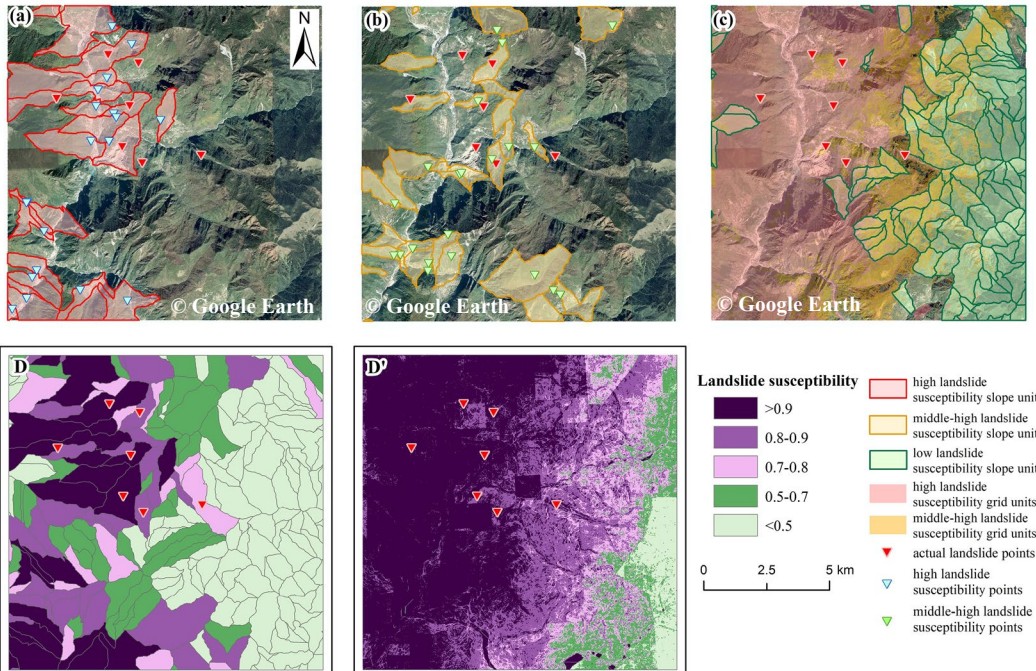

**Figure 5: Comparison of slope units and grid units of quadrat D and landslide susceptibility results in quadrats in remote sensing images. The images come from Google Earth. (a) Possible landslide locations in slope units with high landslide susceptibility and artificial discrimination units; (b) possible landslide locations within medium–high landslide-prone slope units and artificial discrimination units; (c) the range of high and medium–high landslide susceptibilities in the grid unit and the coverage of low landslide susceptibility in the slope unit.**

Figure 6 (I) and (I') are typical quadrats selected from 12 quadrats to demonstrate the underestimation of landslide susceptibility by grid units compared with slope units. The landslide susceptibilities for the grid unit show obvious underestimation. A total of 42.9% of the original landslide points are located in low and medium–low landslide susceptibility areas, while the high landslide susceptibility areas are mainly distributed along the river channel and are not discriminated. The range is small, resulting in areas with measured landslide points also being assessed as low landslide susceptibility areas. Therefore, the landslide susceptibility results based on grid units are not conducive to predicting and preventing geological disaster risks. All the measured landslide points in the slope unit are within the range of high and medium–high landslide susceptibility areas. The prediction of landslide susceptibility in the slope unit is relatively accurate, and the trend of the distribution of each level of susceptibility is reasonable and in line with natural patterns. That is, high and medium–high landslide susceptibility areas are distributed along the river.

Figure 6(a), (b) and (c) to reveal that the slope unit is better than the grid unit in assessing the susceptibility of landslides. Slope unit modelling can more accurately predict the possible occurrence probability of landslides, and for landslides, it also has excellent performance in landslide-prone areas and is not prone to underestimation. Combining the results of high and medium–high landslide susceptibilities in the slope unit and remote sensing images of Google Maps, the possible landslide occurrence points of the unit were visually interpreted. The results are shown in Fig. 6(a) and (b). A total of 70.97% of the



slope units with high landslide susceptibility and 44.12% of the slope units with medium and high landslide susceptibilities have points where landslides may occur. Although only some of the high to medium and high landslide susceptibility ranges

have measured landslide points, combined with satellite remote sensing images, areas where landslides may occur within the high, medium and high landslide susceptibilities of slope units are identified. Therefore, slope units have better predictive performance for high, medium and high landslide susceptibilities. Figure 6(c) shows that the low landslide susceptibility range of the grid unit far exceeds the low landslide susceptibility range of the slope unit, and there are measured landslide points within the low susceptibility range of the grid unit. None of the actual landslide points are within the low landslide-

prone range of the slope unit. Therefore, in this sample case, the slope unit performs better than the grid unit in identifying landslide-prone areas.

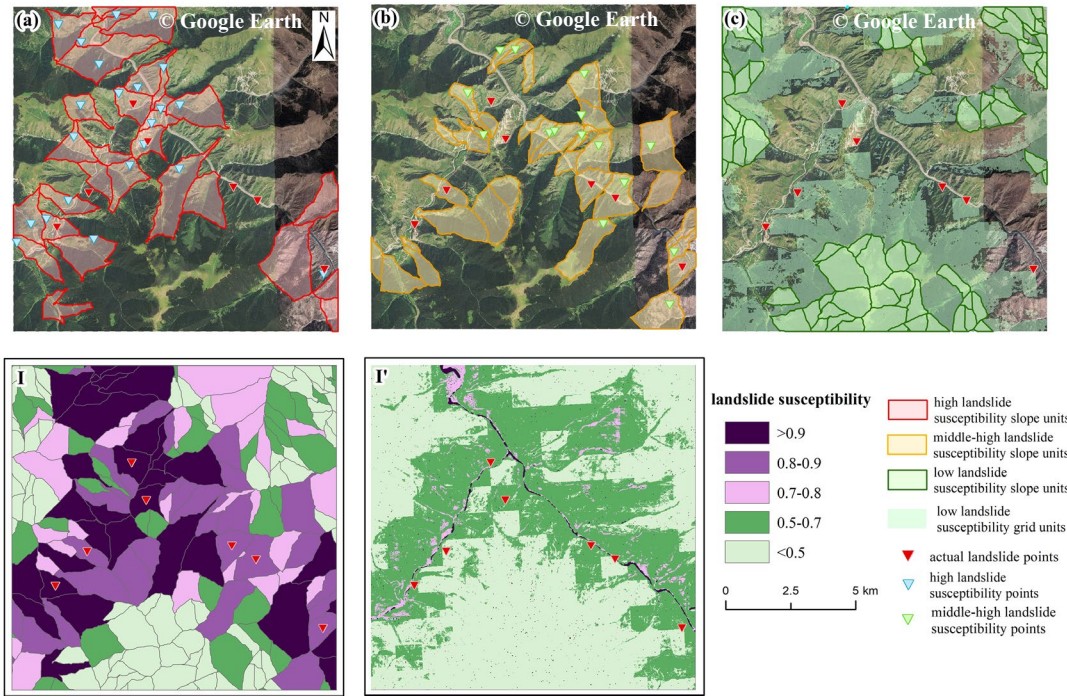

**Figure 6: Diagram comparing slope units and grid units of quadrat I, showing the results of landslide susceptibility in quadrats under remote sensing images. The images come from Google Earth. (a) Possible landslide locations in slope units with high**
**landslide susceptibility and artificial discrimination units; (b) possible landslide locations within medium–high landslide-prone slope units and artificial discrimination units; (c) range of low landslide susceptibilities for the grid unit and range of low landslide susceptibilities for the slope unit.**

## 5.3 Comparison based on the unit scale

One unit from the slope unit quadrat I was selected as the research comparison object at the unit scale, as shown in Fig. 7(a).
This unit was selected because the predicted probability of the landslide susceptibility modelled in this slope unit is 94.59%, which is approximately 27.63% different from the predicted probability of the landslide susceptibility modelled in the grid unit of 66.97%. The gap is large. Based on this actual measurement, the slope unit modelled for the landslide point has a



high landslide susceptibility, which is consistent with the actual situation, but the grid unit has medium and low landslide susceptibilities, which shows an underestimation of landslide susceptibility. Using the selected typical slope units and

combining the influential factors invested in landslide susceptibility modelling, the advantages of slope units compared to grid units at the unit scale are explored. Since this study selected 16 influential factors, in this section, only the factors with larger absolute values of coefficients in the modelling process for the two units are selected, that is, factors that have a greater impact on the model prediction results. Figure 7 (c)-(k) is a raster diagram of influential factors within a single typical slope unit. Table 3 shows the continuous variable coefficient $\beta_i$ of binary logistic modelling for slope units and grid

units in region B. Nine factors with larger absolute values of coefficients were selected, namely, DEM, relative relief, SPI, curvature, distance, river distance, NDVI, slope, TWI, and GDP.

**Table 3.** Coefficients of continuous variables in the binary logistic modelling of slope units and grid units in region B

| factor | value in the slope unit | region B | |
|---|---|---|---|
| | | $\beta_i$ | |
| | | slope unit | grid unit |
| DEM | min | -10.737 | 2.380 |
| Slope | max | 0.872 | -2.868 |
| Aspect | max | 0.048 | 0.561 |
| Relative relief | max | 17.893 | 17.554 |
| Curvature | min | -17.502 | 0.504 |
| Snow | max | -0.410 | -0.582 |
| Temperature | max | -1.546 | -0.567 |
| Rainfall | mean | -0.490 | -0.233 |
| NDVI | min | -2.403 | -0.990 |
| GDP | mean | 9.585 | 2.802 |
| $R_c$ | \ | -0.672 | \ |
| Perimeter | \ | 16.610 | \ |
| Area | \ | 7.533 | \ |
| SPI | mean | 16.835 | 0.545 |
| TWI | mean | 3.183 | -2.193 |
| River distance | mean | -4.719 | -11.428 |
| Fault distance | mean | -0.060 | -0.222 |

For modelling based on grid units, the grid samples were sampled based on the measured landslide points. The measured

landslide points were located in the slope unit with a terrain relief of 26. According to Table 3, the terrain relief was established for the grid unit. The degree of influence in the model is the greatest, but the terrain relief value sampled at this point is relatively small, and the geographical conditions of this landslide hazard point are not prone to landslides according

to existing research results. The raster variable for this point in the prediction result obtained after value modelling is low-risk susceptibility, as shown in Fig. 7 (b). However, actual remote sensing images and measured landslide point datasets prove that this is a highly landslide-prone area, so the point values under the grid unit are not conducive to building a realistic landslide susceptibility model. The value of the influential factors in the slope unit can be used to obtain the maximum, minimum or average value of the grid factors in the slope unit, which can better integrate the differences in values in grid variables in small units and jointly consider the entire geographical conditions of slope units. According to Table 3, terrain relief has the greatest impact in slope unit modelling. The sampling value of terrain relief in the slope unit in this case is 79. According to existing research, slopes with steeper terrain are more likely to cause landslides. The values of the influencing factors raster in the slope unit are more representative, making the coefficient of susceptibility model conform to the regularity of landslide proneness. Therefore, the prediction result of this typical slope unit after modelling is high susceptibility, which is in line with the actual landslide proneness situation.

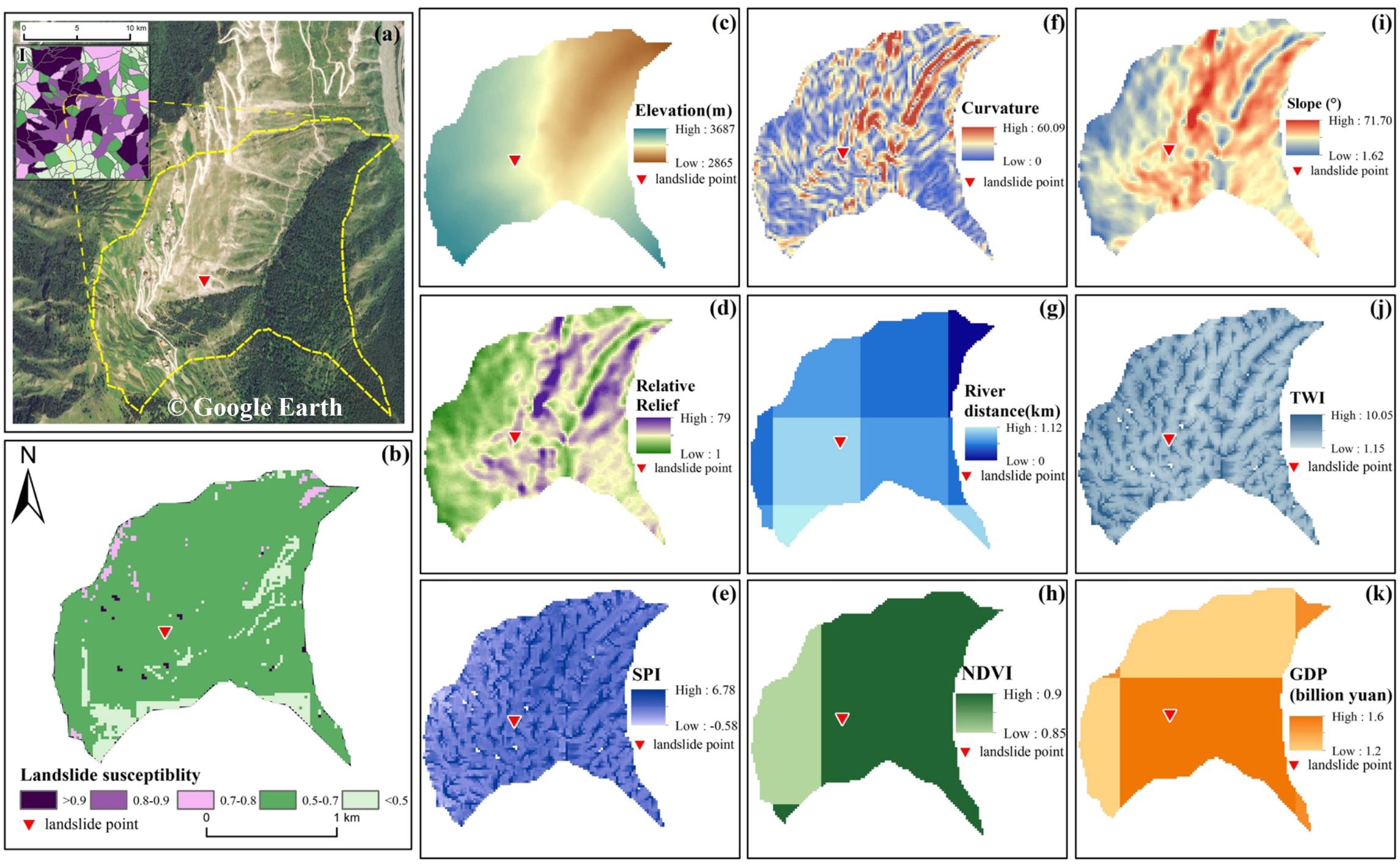

**Figure 7. (a) Enlarged remote sensing image of the slope unit in quadrat I and the images come from Google Earth; (b) map showing the results of grid unit susceptibility in the slope unit; (c)-(k) participants in the landslide susceptibility analysis based on slope units and grid units, respectively. Representative factors in epigenetic modelling, namely, DEM, relative relief, SPI, curvature, distance from the river, NDVI, slope, TWI, GDP and other factors mapped for this typical slope unit.**

Landslides are planar, and the geographical attributes of landslides are regional. Slope units are bounded by watersheds, which can completely divide geological hazards into units. However, automatically divided grid units cannot fully reflect the surface relief state and cannot guarantee landslides. The influential factors that occur are included in the corresponding units.



The original factors are all raster data, and the zoning statistics in the slope unit are the maximum value, minimum value or average value in the area, which can integrate the attributes of the geographical units in the unit and facilitate a more comprehensive analysis and evaluation of landslides in the unit. In the process for grid unit sampling, only the values of
influencing factors at landslide hazard points are taken, and the overall geographical and geomorphological characteristics of landslide hazards are not comprehensively considered. The use of small slope units is more conducive to statistically reasonable variable values and improves the rationality of the model. There is a deviation between the location of the disaster point in the dataset and the actual landslide surface, and it is impossible to accurately locate the starting location of the landslide. However, using small unit data can expand the occurrence of landslides to a geographical unit, thereby lessening
this error.

## 5.4 Verification

Figure 8(a)-(c) shows the ROC curves of the three models in areas A, B, and C based on slope units and grid units in the study area. In Fig. 8(a), the area under the ROC curve (AUC) of the slope unit is 0.914, the AUC of the grid unit is 0.774, and the standard errors are both less than 0.005. In Fig. 8(b), the AUC of the slope unit is 0.904, the AUC of the grid unit is
0.858, and the standard errors are both less than 0.005. In Fig. 8(c), the AUC of the slope unit is 0.814, the AUC of the grid unit is 0.454, and the standard errors are both less than 0.005. In areas A and B, the models based on slope units and grid units have high predictive value; in area C, the models based on slope units have high predictive value. The AUC of the model under the grid unit is less than 0.5, and the prediction results are not credible. Within the study area, the predictive value of the model based on slope unit modelling is better than that of grid unit modelling, indicating that slope units have
superior                    performance                 in               building              landslide                susceptibility               models.

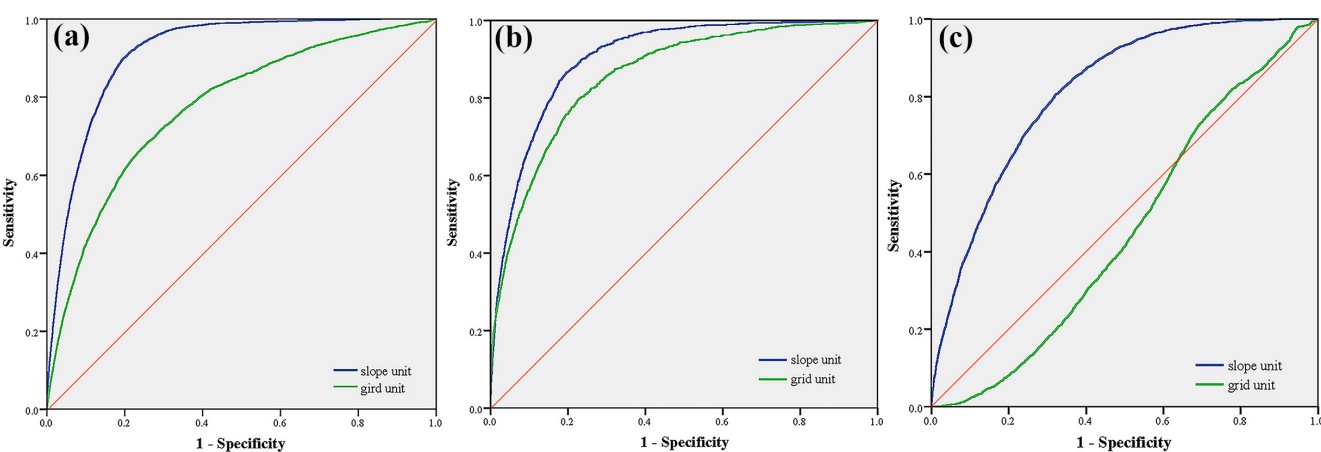

**Figure 8. ROC curve for the model based on the slope and grid units. (a) region A, (b) region B, (c) region C.**



## 5.5 Limitation

While our slope unit dataset provides valuable map units for evaluating geohazard mapping units, it has limitations that must
be considered. When extracting slope units from DEM data, hydrology tools of ArcGIS delineated error units in flat areas,
forming parallel horizontal lines. Therefore, this dataset was manually corrected for waters and flat areas. So, the dataset
inevitably contains subjectivity from our inspectors in the above areas. And because ridge lines and valley lines do not exist
over water surface, the dataset does not contain slope units at lakes.

Since the slope unit in the dataset was extracted from 12.5m DEM data. When assessing geological hazards, the spatial
resolution of the influencing factor data put into is preferably higher than 1 km. And resampling is needed to improve the
spatial resolution of factor data and facilitate slope unit sampling.

## 6 Data availability

The slope unit data of the south-eastern edge of the Tibetan Plateau and associated data (the landslide susceptibility
probability, the area and perimeter of the slope units and the existence of actual landslides) are available from the repository
hosted on Figshare: 10.6084/m9.figshare.24457144 (Zheng et al.,2023). Data are provided as shapefile files (.shp).

## 7 Conclusions

This study uses water system extraction and sub-basin segmentation methods to obtain slope units in the mountainous areas
of south-eastern Tibet with a DEM with a higher spatial resolution of 12.5 m and manually edits abnormal slope units in the
entire study area with reference to the shaded terrain map. This resulted in the provision of slope units covering an area of
350,000 square kilometres with a unit number of 446,497. Sixteen natural factors that affect landslides were selected,
including DEM, aspect, slope, relief, curvature, SPI, TWI, distance from rivers, distance from fault zones, snow cover,
precipitation, temperature, NDVI, GDP, land use, and lithology. Social factors are modelled using binary logistic equations
based on slope units and grid units, and landslide susceptibility maps with slope units and with grid units with 12.5 m spatial
resolution are obtained. The AUC values of the landslide susceptibility models with slope units are all greater than those of
grid units, with AUC values of 0.814-0.914 and standard errors less than 0.005, which shows that slope units are better than
the widely used grid units for predicting landslide susceptibility.

According to the landslide susceptibility map for the slope unit, there are approximately 13,059 highly susceptible slope
units. The area with high landslide susceptibility in the mountainous area of south-eastern Tibet is approximately 18,044.58
square kilometres, accounting for 5.20% of the total area. Units with high landslide susceptibility tend to be distributed along
rivers, mainly along the Yalong River, Min River and Jinsha River. Due to the active geological structures in the terrain
transition zone, a large number of units with high landslide susceptibility are also concentrated in the transition area between
the Sichuan Basin and the Western Sichuan Plateau. The trend of the distribution of highly landslide-prone areas in slope



units is similar to that of actual landslide hazard areas. However, the trend of the distribution of highly landslide-prone areas in grid units is obviously different from that of actual landslide areas, and there is a phenomenon of overestimation or underestimation.

At higher spatial resolutions, the accuracy and predictive performance of landslide susceptibility modelling based on slope units are better than those of grid units. This study randomly selected 12 quadrats and explained these quadrats at three scales: overall scale, quadrat scale, and unit scale. Slope units are superior to grid units as landslide susceptibility evaluation units. At the overall scale, the trend of the distribution of landslide susceptibility in the slope unit is more consistent with the landslide occurrence pattern; that is, highly susceptible landslide areas are mainly distributed along rivers, and the modelling accuracy of slope units is better than that of grid units with high spatial resolution. At the quadrat scale, compared with grid units, slope units are less likely to underestimate landslide-prone areas and are less likely to overestimate landslide-prone areas, and the landslide susceptibility of slope units is smaller. The results can more accurately predict hidden danger points where landslides may occur. At the unit scale, a single slope unit can comprehensively evaluate the factors influencing landslides within the unit range and consider the overall physical and geographical attributes within the units. It is conducive to statistics of influencing factors variables that are more in line with landslide occurrence, thereby improving the accuracy of landslide susceptibility model. Deviation between the location of the disaster point in the dataset and the actual landslide surface occurs, and it is impossible to accurately locate the starting location of the landslide. However, using few unit data points can expand the occurrence of landslides to a geographical unit, thereby weakening this error. Therefore, this slope unit dataset provides fine slope unit vector data and the number of landslide hazard points in the eastern edge of Tibetan Plateau, which is beneficial for carrying out research related to geological hazards in this region.

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
