# Peer review of "A geomorphological slope unit dataset for the eastern edge of Tibetan Plateau"

_Earth System Science Data, 2023_

## Author Comment (AC1)

We would like to thank the referee for the time and effort they put in to review the first version of our manuscript. Their constructive comments enabled us to improve the quality and clarity of the manuscript. We have checked the grammar and spelling errors of the manuscript. Please find our answers to the points raised below.

Q1: Please specify how "slope units" differ from "watersheds" and, if they are different, please provide a better definition for slope units.

A1: Thank you for your comments. Actually, the definition of our units in this study is similar with the watersheds, but the area of our units is smaller than the general watershed. So, we redefined our units as geomorphic units. Please see **Line 26-30:**

Geomorphic units are irregular areas delineated by ridge lines and reflect comprehensive landform characteristics at a small scale (Tang and Ma, 2015). Geomorphic units are also different from watersheds because they do not necessarily include the whole hydrological system. Moreover, the area range of general watershed are more than 100 km$^2$(Xu et al., 2004), but geomorphic units of this study are smaller than 5 km$^2$, which are more applicable to landslide susceptibility assessment.

Tang, C, Ma, C. Small Regional Geohazards Susceptibility Mapping Based on Geomorphic Unit, Scientia Geographical Sinica, 35:1, 92-98, DOI: 1000-0690(2015)01-0091-08, 2015.
Xu, X, Zhuang, D, Jia, S, et al. Automated Extraction of Drainages in China Based on DEM in GIS environment. Resources and Environment in the Yangtze Basin, 13(4):343-348, 1004-822 (2004) 04-0343-06, 2004.

Q2: Figure 5 and 6 and all the discussion concerning the better performance of slope units compared to grid units is based on the actual landslide points and some other landslide points of which the origin is not explained – or not sufficiently explained. This makes it difficult to understand the real performance of this method.

A2: Thank you for your comment. We have explained how the google images to identify possible landslide points. Please see **Lines 280-320:**

[Figure]

**Figure 5: Comparison of geomorphic units and grid units of quadrat D and landslide susceptibility results with actual landslide point and remote sensing interpretation landslide points. The RS images come from Google Earth.**

**(a) the coverage of high landslide susceptibility in the geomorphic unit; (b) the coverage of middle-high landslide susceptibility; (c) the coverage of low landslide susceptibility in the geomorphic unit.**

The dark red points in Fig. 5 and Fig. 6 are the landslide point, which are obtained from manually interpreted the Google Earth remote sensing images. According to the Remote sensing image, these landslide points are mostly the bare landform. There are dense vegetation around these points. The locations of these possible landslide points are within the centers of the potential landslide areas. Red points in Fig. 5 and Fig. 6 come from "The standard specification of geological hazard risk survey and assessment in China", which obtained from the fieldwork. These actual observation landslide points are concerned that may occur and are potentially dangerous to surrounding homes, roads, etc. The difference between dark red points and red points is that dark red points are less dangerous to houses and roads, so there are no dark red points in the database.

In this paper, Fig. 5(a), (b) and (c) show that the geomorphic unit method is better than the grid unit method in determining landslide susceptibility, and the geomorphic unit method can predict the probability of landslides more accurately and has excellent performance in non-landslide areas. A total of 81.82% of geomorphic units with high landslide susceptibilities and 76.92% of geomorphic units with medium and high landslide susceptibilities have possible landslide points. Although some of the geomorphic units with high and medium landslide susceptibilities do not have actual landslide points (red points), the possible landslide points (dark red points) within the unit range can be interpreted in satellite remote sensing images. Therefore, the geomorphic unit has better performance in identifying high and medium-high landslide susceptibilities. Based on the data presented in Figure 5(c), no landslide point is located within the low-landslide susceptibility geomorphic units. Therefore, in this sample case, the geomorphic unit method performs better than the grid unit method in identifying landslide-prone areas.

[Figure]

**Figure 6: Comparison of geomorphic units and grid units of quadrat I and landslide susceptibility results with actual landslide point and remote sensing interpretation landslide points. The RS images come from Google Earth. (a) the coverage of high landslide susceptibility in the geomorphic unit; (b)**

**the coverage of middle-high landslide susceptibility; (c) the coverage of low landslide susceptibility in the geomorphic unit.**

Figure 6 (I) and (I') are typical quadrats selected from 12 quadrats to demonstrate the underestimation of landslide susceptibility by grid units compared with geomorphic units. A total of 98% of the actual landslide points are located in low and medium–low landslide susceptibility areas shown in the Fig.6 (I').

Figure 6(a), (b) and (c) reveal that geomorphic units are better than grid units in assessing the susceptibility to landslides. Geomorphic unit modeling can more accurately predict the possible occurrence probability of landslides, and for landslides, it also has excellent performance in landslide-prone areas and is not prone to underestimation. The results are shown in Fig. 6(a) and (b). A total of 70.97% of the geomorphic units with high landslide susceptibility and 44.12% of the geomorphic units with medium and high landslide susceptibilities have actual landslide points (red points), and some of the medium-high and high landslide susceptibility ranges have interpreted landslide points (dark red points). Figure 6(c) shows that none of the actual landslide points are within the low landslide susceptibility range based on the geomorphic unit method. Therefore, in this sample case, the geomorphic unit method performs better than the grid unit method in identifying landslide-prone areas.

There are two reasons why geomorphic units are better at assessing landslide susceptibility than grid units are. Above all, geomorphic units have geomorphic characteristics and reflect the impact factors of landslides during sampling. Second, the susceptibility of geomorphic units can be assessed according to the comprehensive characteristics of influencing factors, while the susceptibility of grid units can be easily controlled by a single influencing factor.

Q3: Lines 18-19: "To enable more researchers to focus more conveniently on the subject matter to be addressed itself, rather than being caught up in the slope unit delineation." This sentence is not grammatically correct. Please check.
A3: Thank you for your comments. We have checked the grammar of this sentence. Please see **Line 16-18:**

This study enables more researchers to focus on hazard susceptibility assessments based on geomorphic units rather than the complicated process of geomorphic unit delineation.

Q4: Line 28: slope units are the area**s** …
A4: Thank you for your comments. We have corrected the grammar. Please see **Line 27:**
Geomorphic units are irregular areas delineated by ridge lines …

Q5: Lines 28-29: please rephrase, what does it mean "the basic topographic units of natural geological hazards"? not clear.
A5: Thank you for your comments. We have rephrased the sentence. Please see **Line 27-28:**
Geomorphic units are irregular areas delineated by ridge lines and reflect comprehensive landform characteristics at a small scale (Tang and Ma, 2015), …

Q6: Line 29: hazard evolution? Do you mean, hazard evaluation? If not, please specify what hazard evolution means for you – based on climatic changes? Naturally evolve in time? Not clear.

A6: Thank you for your reminder. We mean "hazard assessment" and correct it. Please see **Line 30-31:**

Geomorphic units are usually used in hazard assessment, such as landslide susceptibility, debris flow risk and flood risk,

Q7: Lines 30-31: this sentence is odd: possibly change to "Due to the intense tectonic activity and complex topography of the eastern edge of Tibetan Plateau, steep slopes are prone to deformation and failure". Please check.

A7: Thank you for your comment. We have checked this sentence and change as your suggestion. Please see **Line 56-58:**

Due to the intense tectonic activity and complex topography of the eastern edge of the Tibetan Plateau, steep slopes are prone to deformation and failure (Dai et al., 2019, Zhao et al., 2021; Zhan et al., 2018).

Q8: Line 34: "the mapping units regarded as the sampling units" not clear. What do you mean with sampling unit, the areas normally used to perform topographic analysis? Please clarify.

A8: Thank you for your comment. We have used another description namely "analysis units". Please see **Line 32-34:**

In the field of geologic hazard research, analysis units, such as grid units (Du et al.,2019), slope units (Hua et al.,2021), unique-condition units (Domenico et al.,2010) and geomorphic units (Qin et al., 2019), are diverse and are used to perform geographical analysis (Rotigliano et al., 2012).

Q9: Line 37: what do you mean with "geological environments"? if the grid units are basically the areas of a single grid cell, I would say that the biggest problem here is the raster resolution. Are we talking about DEMs? Then, if a grid cell represents an area of 30x30m, of course it will average all topographic features over a large area. If resolution is of few meters, then the definition of topographic features can have high spatial definition.

A9: Thank you for your comment. We apology for the unclear statement. The following sentence from this article is explained our meaning.

"Drawbacks lay in the absence of any relation between grid-cells and geological, geomorphological, or any other terrain information." (Guzzetti et al. 1999)

In addition, your perspective is correct, but as the following paper saying:

"The tendency to use smaller and smaller grid-cells appears unjustified. Spatial inaccuracy is partially reduced but to cover even small areas an overwhelming number of grid-cells is required, leading to unmanageable computer problems and numerical instability when data have to be processed by statistical techniques." (Guzzetti et al. 1999)

*Guzzetti F, Carrara A, Cardinali M, Reichenbach P. Landslide hazard evaluation: a review of current techniques and their application in a multi-scale study central Italy, J. Geomorphology 31(1-4): 181–216.https://doi.org/10.1016/S0169-555X(99)00078-1, 1999.*

Q10: Lines 41-42: I still did not get how slope units are defined, and what is meant with geological environments. Besides the benefits of using slope units, it would be nice to also know the disadvantages. I guess that raster resolution still remains an important limitation.
A10: Thank you for your comment. We apology for the unclear statement and delete relating sentence. Please see **Line 26-30.**

Q11: Lines 43-44: this sentence is written in a strange English, please rephrase.
A11: Thank you for your comment. We have rephrased this sentence. Please see **Line 37-38:**
Some studies have explored efficient and objective methods for delineating slope units due to their high frequency in natural hazard assessments.

Q12: Line 45: please add, GIS software.
A12: Thank you for your comment. We have added the "GIS software" in this sentence. Please see **Lines 38-40:**
Initially, the geomorphic units were delineated manually by GIS software based on topographic maps, which is subjective and time-consuming and limits the application area (Carrara, 1988; Alvioli et al., 2016).

Q13: Line 46: Hydrological tools are available for several GIS software, not only for ArcGIS. Additionally, these tools are those generally used to define watersheds, and I still did not get how slope units differ from them.
A13: Thank you for your comment. We have changed the "Arcgis" to "GIS software" in this sentence. Please see Line 47-48. And the difference between the slope units and watersheds units have explained in the A1. **Lines 40-41:**
With the development of GIS, some studies have utilized the hydrology tools of GIS software to extract ridge lines for generating geomorphic units (Xie et al., 2004; Erener and Düzgün 2012).

Q14: Line 47: This sentence is written in an awkward English. Please check. Also, "r.slopeunits" seems a function of the R software, or one implemented in QGIS. Please, remove the word software after it.
A14: Thank you for your comment. We have removed the word "software" and rewritten this sentence. Please see **Lines 42-43:**
An automatic method based on r.slopeunits can be used to delineate more reliable and objective evaluation units (Alvioli et al., 2016).

Q15: Line 48: which new method?
A15: Thank you for your comment. We have explained the new method in this sentence. Please see **Lines 43-44:**
Another new method, derived from morphological image analysis with logical algorithms, can extract more homogeneous geomorphic units (Wang et al., 2019).

Q16: Line 49-50: how is, having different sources and resolution, a problem for slope unit extractions?
A16: Thank you for your comment. We have clarified the problem for slope unit extractions used difference sources and resolution of DEM. Please see **Lines 44-45:**

Geomorphic units can be delineated from different DEM spatial resolutions, which impacts their size and form (Tian et al.,2019).

Q17: Line 51-52: this sentence is awkward, with repetitions. Please rephrase.
A17: Thank you for your comment. We have rephrased this sentence. Please see **Lines 49-50:**
The eastern edge of the Tibetan Plateau has large areas prone to geological hazards, where hazard assessment research needs to be carried out (Du et al.,2017).

Q18: Lines 43-54: I understand the intention of the authors, to provide a background for the utility of their work, but this paragraph is written in a very poor English, and it is really hard to understand. I strongly recommend rewriting it.
A18: Thank you for your comment. We have rewritten this paragraph. Please see **Lines 37-48:**
Some studies have explored efficient and objective methods for delineating slope units due to their high frequency in natural hazard assessments. Moreover, the delineation methods for slope units are also suitable for geomorphic units. Initially, the geomorphic units were delineated manually by GIS software based on topographic maps, which is subjective and time-consuming and limits the application area (Carrara, 1988; Alvioli et al., 2016). With the development of GIS, some studies have utilized the hydrology tools of GIS software to extract ridge lines for generating geomorphic units (Xie et al., 2004; Erener and Düzgün 2012). An automatic method based on r.slopeunits can be used to delineate more reliable and objective evaluation units (Alvioli et al., 2016). Another new method, derived from morphological image analysis with logical algorithms, can extract more homogeneous geomorphic units (Wang et al., 2019). Geomorphic units can be delineated from different DEM spatial resolutions, which impacts their size and form (Tian et al.,2019). However, the geomorphic units do not have a unified standard. Moreover, the existing studies on extracting geomorphic units typically cover only a river basin, which does not provide a sufficient dataset for other studies. Therefore, geological hazard assessment requires comprehensive coverage of geomorphic unit datasets with a unified extraction standard (Guzzetti et al. 2005).

Q19: Lines 55-60: please write areas as plural. Line 58: DEM with low resolution – not small scale. Resolution and scale are different things. Line 60: areas as plural. Please here, and throughout the paper, avoid using geological disaster. Use instead geological hazards.
Q19: Thank you for your comment. We have written areas as plural, changed the scale to resolution and used the geological hazards throughout the paper. Please see **Lines 49-57:**
The eastern edge of the Tibetan Plateau has large areas prone to geological hazards, where hazard assessment research needs to be carried out (Du et al.,2017). Existing research has assessed and mapped landslide susceptibility based on grid units in the part region of Tibetan Plateau (Wang et al.,2017; Sun et al.,2020). However, few studies have assessed hazard susceptibility based on geomorphic units at large areas. Due to the intense tectonic activity and complex topography of the eastern edge of the Tibetan Plateau, steep slopes are prone to deformation and failure (Zhan et al., 2018; Dai et al., 2019, Zhao et al., 2021). When extracting units from DEMs, the study areas have been limited to only small areas, such as watershed (Li & Lan et al.,2020; Wang et al.,2020). For large study areas, the

geomorphic units mostly extracted from 15m or 30m DEMs (Gregory C. O., & John C. D., 2003; Tian, S. & Kong, J.,2013; Liu et al., 2021). Accordingly, few geomorphic unit datasets cover large areas for the assessment of geological hazards (Chung, C.-C. & Li, Z.-Y, 2022).

Q20: Lines 62-63: please rewrite – this sentence is repetitive and odd. How would your new database overcome the limitations for their application in small areas?

A20: Thank you for your comment. We have rewritten this sentence. Please see **Lines 58-59:**

This study constructed a geomorphic unit database for the eastern edge of the Tibetan Plateau.

Q21: Line 65: is this resolution a high resolution compared to data normally used for the Tibetan plateau?

A21: Yes, it is. Thank you for your comment. We have deleted this "high- resolution" in this paragraph. Please see **Lines 61-66:**

In this study, based on 12.5 m DEM data and GIS software hydrology tools, geomorphic units on the eastern edge of the Tibetan Plateau were extracted. Based on the 12.5 m grid units and the geomorphic units in this dataset, the landslide susceptibility of the eastern edge of the Tibetan Plateau was assessed with a logistic model. Then, the landslide susceptibility results of the two kinds of units were compared based on the overall , sample , and geomorphic unit. Moreover, this study analyzed the influencing landslide factors among two kinds of units. This dataset is expected to contribute to future geological hazard research in the hazard-prone area of the eastern edge of the Tibetan Plateau.

Q22: Lines 66-67: "And the error part of slope units ??? was manually modified which were delineated from the water surface ???". I do not understand this sentence. What are you talking about? Shouldn't this go in the method section?

A22: Thank you for your comment. We have rewritten this sentence in the method section to facilitate your understanding. Please see **Lines 193-194:**

Therefore, the geomorphic units in flat regions in the study area were difficult to delineate correctly via GIS software, and we manually delineated the geomorphic units in flat regions according to hill shade images.

Q23: Line 81: what is the vertical and horizontal distribution of rivers?

A23: Thank you for your comment. We have rewritten this sentence for more easily to understand. Please see **Lines 77-79:**

The crisscrossing and dense distributions of rivers have produced intense river erosion and deep canyon landforms, with relative elevations greater than 1000 m accounting for 81.5% of the total area (Li, 1989; Bian et al., 2018).

Q24: Lines 82-83: the monsoon is the main air mass for precipitation transport.

A24: Thank you for your comment. We have revised this sentence according to your suggestion. Please see **Lines 84-85:**

This study region is mainly influenced by the Northern Hemisphere westerly circulation and monsoons, and the monsoon is the main air mass for precipitation transport (Wei et al., 2018).

Q25: Line 97: remove DEM from main DEM data. DEM stays for Digital Elevation Model, here it is intended as RASTER.

A25: Thank you for your comment. We have revised this sentence according to your suggestion. Please see **Lines 96-97:**

The main data used in this study are 12.5 m DEM data collected by phased array L-band synthetic aperture radar (PALSAR) from the Advanced Land Observing Satellite (ALOS);

Q26: Line 100: what kind of information provide the GDP?

A26: Thank you for your comment. The information of GDP provided is the intensity of human activity. Please see **Lines 140-142:**

GDP has a strong correlation with human activity intensity (Peng et al., 2021). Accordingly, this study selected the annual mean GDP as the landslide factor indicating human activity intensity.

Q27: Line 103: are freely available, or are available for free

A27: Thank you for your comment. Those data are available for free. And we have revised this sentence. Please see **Line 114:**

All the above datasets are available for free at the URLs listed in Table 1.

Q28: Line 111: what do you mean with geographical environment, can't you just say, within the study area?

A28: Thank you for your comment. The "geographical environment" means the geographical condition. And we have revised the sentence according to your suggestion. Please see **Line 109:**

Within the study area, 16 factors influencing landslides were selected, …

Q29: Line 114: DEMs is plural, which **are** indicative …

A29: Thank you for your comment. We have revised this sentence according to your suggestion. Please see **Lines 116-117:**

DEMs are the most commonly used datasets in landslide susceptibility analysis and are indicative of topography and landform morphology.

Q30: Line 115: with level? Better to specify, with the horizon, or horizontal level

A30: Thank you for your comment. We have revised this sentence according to your suggestion. Please see **Line 117:**

The slope is the angle formed by any earth surface with the horizontal level, …

Q31: Line 116: please specify what aspect is. Not only which values it may have.

A31: Thank you for your comment. We have specified what aspect is. Please see **Lines 118-120:**

The aspect is the direction of the slope normal projected to the horizontal level, namely, the downhill direction of the slope, which ranges from 0° to 360°. northern slope. Aspect class as flat (–1°), north (315°–360°, 0°–45°), east (45°–135°), south (135°–225°), and west (225°–315°)

Q32: Line 117: you mean, the aspect, not the slope. How is precipitation influenced by the aspect (or the slope)?

A32: Thank you for your comment. Existing papers have studied the correlation between the rainfall and aspect. Please see **Lines 120-124:**

Some of the meteorologic events such as the direction of the rain, amount of sunshine, the morphologic structure of the area which that examined affected the propensity for landslides to intensify in terms of the type of the slope. Aspect is related to the parameters that can control the formation of the landslide such as lineaments, rainfalls, wind effects, and exposure to sunshine. That landslides were frequently observed in some aspect values was determined by statistical evaluation in most of the studies (Yalcin and Bulut, 2007).

*Yalcin, A., Bulut, F. Landslide susceptibility mapping using GIS and digital photogrammetric techniques: a case study from Ardesen (NE-Turkey). Nat Hazards 41, 201–226. https://doi.org/10.1007/s11069-006-9030-0,2007.*

Q33: Line 118: the forces of slope material ?? what is it? You mean cohesion? Please rephrase

A33: Thank you for your comment. Actually, we mean cohesion of slope block and have rephrased the expression. Please see **Lines 126:**

Curvature affects the cohesion of the slope block and the movement of runoff.

Q34: Line 123: active faults may be a triggering factor – please rephrase; Line 124: which same way?

A34: Thank you for your comment. We have rephrased this sentence and explained the same way. Please see **Lines 132-133:**

Active faults may be triggering factors for landslides, so the distance to a fault is also calculated as the Euclidean distance with GIS software.

Q35: Line 126: "The mean annual temperature data are represented by the temperature index" what does it mean?

A35: Thank you for your comment. We have rephrased this sentence to easily understand. Please see **Lines 134:**

The mean annual temperature data are selected as the temperature factor.

Q36: Line 127: what are the unstable structure in rocks? Do you mean faults?

A36: Thank you for your comment. We have rephrased this sentence to explain. Please see **Lines 135-136:**

Lithology reflects the characteristics of rocks, and some kinds of rocks are more prone to landslides.

Q37: Line 128: "Rainfall is usually the factor inducing landslides" it may be, or at least provide a reference for this.

A37: Thank you for your comment. We have added references in this sentence. Please see **Lines 136-137:**

Rainfall is usually the factor inducing landslides (Segoni, S., et al. 2018; Guzzetti, F. et al., 2022), so the average annual rainfall is set as the susceptibility factor.

*Segoni, S., Piciullo, L. & Gariano, S.L. A review of the recent literature on rainfall thresholds for landslide occurrence. Landslides 15, 1483–1501. https://doi.org/10.1007/s10346-018-0966-4, 2018.*

*Fausto Guzzetti, Stefano Luigi Gariano, Silvia Peruccacci, Maria Teresa Brunetti, Massimo Melillo, Chapter 15 - Rainfall and landslide initiation, Pages 427-450, Rainfall, Italy, https://doi.org/10.1016/B978-0-12-822544-8.00012-3, 2022.*

Q38: Line 130: please explain what GDP is – more precisely.

A38: Thank you for your comment. We have added references in this sentence. **Please see Lines 139-141:**

GDP is the market value of the results produced by economic activities in a region within a given period and is a commonly used index to measure economic development (Zhang et al., 2022). GDP has a strong correlation with human activity intensity (Peng et al., 2021). Accordingly, this study selected the annual mean GDP as the landslide factor indicating human activity intensity.

*Zhang T, Sun Y, Guan M, Kang J, et al. Human Activity Intensity in China under Multi-Factor Interactions: Spatiotemporal Characteristics and Influencing Factors. Sustainability. 14(5):3113. https://doi.org/10.3390/su14053113, 2022.*

*Peng K, Zhang Y, Gao W, et al. Evaluation of human activity intensity in geological environment problems of Ji'nan City, European Journal of Remote Sensing, 54:sup2, 117-121, DOI: 10.1080/22797254.2020.1771214, 2021.*

Q39: Line 152: do you mean "slope units"?

A39: Thank you for your comment. We mean geomorphic units and have rewritten in the sentence. Please see **Line 163-164:**

Since the number of geomorphic units with landslide hazards is 9579 and the number of geomorphic units without landslide hazards is 441159, …

Q40: Lines 152-153: "landslide events account for 2.17% of non-landslide events, which can be considered rare events." not clear. Maybe rephrase, "landslide events are found in only the 2.17% of the total slope units, and can thus be considered as rare events" (if I have understood correctly what you wrote)

A40: Thank you for your comment. We have rephrased according to your suggestion and you actually understood correctly what we wrote. Please see **Lines 163-164:**

Since the number of geomorphic units with landslide hazards is 9579 and the number of geomorphic units without landslide hazards is 441159, geomorphic units with landslide are found in only 2.17% of the total geomorphic units.

Q41: Lines 152-156: I am sorry, I do not understand what you are doing here. Please add description and details. Please see **Lines 161-168:**

A41: Thank you for your comment. We have rephrased this paragraph and added description and details.

Since the number of geomorphic units with landslide hazards is 9579 and the number of geomorphic units without landslide hazards is 441159, geomorphic units with landslide are found in only 2.17% of the total geomorphic units. Then, we treated the geomorphic units with landslides as "1"; otherwise, they were treated as "0". During the application of the logistic regression model, fewer "1" samples grossly underestimate the probability of

landslide occurrence (King & Zeng, 2001). Therefore, keeping the ratio of "0" samples to "1" samples from 0.2 to 1 is beneficial for precisely predicting the probability of landslides (Basu, T. & Pal. S, 2017). Therefore, this study copied the "1" geomorphic units to increase their number for 46 times, The number of "0" to "1" geomorphic units is approximately equal (Wang et al., 2016; Wang et al, 2018). This oversampling method can realize a balance among the training samples (Yen et al., 2006).

*King G, Zeng L, 2001. Logistic regression in rare events data. Political Analysis, 9(2): 137–163.*
*Yen S J, Lee Y S, Lin C H et al., 2006. Investigating the effect of sampling methods for imbalanced data distributions. In: Systems, Man and Cybernetics, 2006. SMC'06. IEEE International Conference, 5: 4163–4168.*
*Wang, Y., Song,C., Lin, Q, et al. Occurrence probability assessment of earthquake-triggered landslides with Newmark displacement values and logistic regression: The Wenchuan earthquake,China,Geomorphology,258:108-119, https://doi.org/10.1016/j.geomorph.2016.01.004, 2016.*
*Wang, Y., Lin, Q. & Shi, P. Spatial pattern and influencing factors of landslide casualty events. J. Geogr. Sci. 28, 259–274 (2018). https://doi.org/10.1007/s11442-018-1471-3*

Q42: Lines 157-160: this is not clear. Please rephrase. What span, those of the model? Write down the correspondence between degree and m2. What is referred to as geographical environment? Please clarify. What are the "internal and external forces of the landform"??

A42: Thank you for your comment. We have rephrased these sentences and clarified these phrases. The geographical environment is referred to as geographical feature. The "internal and external forces of the landform" means endogenetic and exogenic processes. Please see **Lines 171-174:**

In this study, due to the large extent of the study area, the latitudinal span is approximately 834.8 km, and the longitudinal span is approximately 922.7 km. Large differences exist in geographical features, such as precipitation, temperament, lithology and so on, so the overall modeling accuracy is low. Therefore, according to the characteristics of landforms, landslide susceptibility modeling is carried out by dividing the study area into three parts, namely, regions A, B, and C. The landforms in A are mainly extremely high mountains and high valleys, those in B are mainly alpine valleys, and those in C are mainly high mountains and plateaus

Q43: Line 163: isn't this still part of the method?

A43: Thank you for your comment. Yes, this chapter is still the part of method and we have moved it to the method. Please see **Line 177:**

3.4 Generation of geomorphic units

Q44: Line 165: unique-conditions units, were not described as "administrative units" in the introduction – please be consistent.

A44: Thank you for your comment. We have deleted the "administrative units" to be consistent with the introduction. Please see **Lines 178-179:**

In landslide susceptibility analysis, commonly used basic assessment units include grid units, slope units, geomorphic units, and unique condition units.

Q45: Line 169: "limiting by the geographic irrelevance." what does it mean?

A45: Thank you for your comment. We have explained it in the sentence. Please see **Lines 398-399:**

However, the landslide susceptibility estimates based on grid units are inconsistent with the actual landslide distribution pattern.

Q46: Line 170: terrain trends, do you mean terrain features?

A46: Thank you for your comment. Yes, we mean terrain features and have rephrased in this sentence. Please see **Lines 181-182:**

When evaluating landslide susceptibility, using geomorphic units is more consistent with the geographical distribution of landslides, and the susceptibility results also attain more terrain features than grid units.

Q47: Lines 168-171: but you can say this after you have shown your results, not now! This is something that should go in the discussion.

A47: Thank you for your comment. We have moved it to the discussion. Please see **Lines 389-392:**

The high usage rate of grid units is because most geographical factor data are stored in raster form, so they are easy to process at various resolutions and scales. However, the landslide susceptibility estimates based on grid units are inconsistent with the actual landslide distribution pattern. Therefore, this study provides a geomorphic unit dataset to assess landslide susceptibility conveniently and precisely.

Q48: Line 172: what is "water system extraction"?

A48: Thank you for your comment. "Water system extraction" actually means "ridge line extraction". We have rewritten in this sentence. Please see **Line 183:**

According to catchment principle, this dataset uses DEM data for the extraction of ridge lines to segment geomorphic units.

Q49: Line 174: "Valley line and ridge line layers are superimposed to obtain slope units" How?

A49: Thank you for your comment. We apologize for confusing the slope units and geomorphic units. Geomorphic units are irregular areas delineated by ridge lines and reflect comprehensive landform characteristics at a small scale. So, we don't use the valley line. We have added the detail in this sentence. Please see **Line 184-185:**

The ridge line layers were superimposed to obtain geomorphic units via the merge tool of GIS software.

Q50: Line 175-176: no, you use the fill depression function to allow water to run out of any real or fake depression, otherwise the other GIS functions will not work.

Q51: Line 177: "as the flow rate"?? flow accumulation is not the same thing as flow rate

Q52: Lines 177-179: this is a repetition, please simplify.

Q53: Line 179: decomposed??? Please change word.

Q54: Line 180-181: it is not clear what is meant by flow rate. Also, change catchment depression with catchment outlet or confluence. It is not clear what catchment depression is.

Q55: Line 182: isn't the valley line the same thing as the river? Not clear.

Q56: Line 183. Still it is not clear how slope units are defined and created.

A50-56: Thank you for your comment. We apologize for confusing the slope units and geomorphic units. We have rewritten the process of generating the geomorphic units. Please see **Line 197-203:**

First, depressions in the DEM data were filled by a fill depression function to allow water to run out of any real or fake depression. The flow direction was calculated based on the elevation after filling, and then the upstream catchment area was calculated based on the flow direction as the flow accumulation. A river network was formed after the flow reached a certain threshold, which was set by the number of upstream water collection units. The river network was segmented into branches according to nodes, which serve as local confluence depressions. With reference to flow accumulation, it got from the watersheds, which the boundaries of watershed are the ridge lines. Finally, the ridge line was used to form geomorphic units.

Q57: Line 198: landslide occurrence probability?

A57: Thank you for your comment. Yes. We have changed the word in this chapter.

Q58: Lines 197-206: please rephrase, or simply add this data in a table. Lines 220-230: report data in a table.

A58: Thank you for your comment. We have added a table to report data and rephrased this paragraph. Please see **Lines 206-210:**

The number and proportion of slope and grid units under each landslide susceptibility probability are shown in Table 2. This result indicates that the predicted number of geomorphic units with high landslide susceptibility is more accurate than the predicted number of grid units.

**Table 2.** The number and proportion of slope and grid units within each landslide susceptibility probability range

| probability of landslide susceptibility | landslide susceptibility level | number of geomorphic units | proportion (%) | number of grid units | proportion (%) |
|---|---|---|---|---|---|
| >0.9 | high | 13,059 | 2.90% | 177,217,634 | 8.80% |
| 0.8-0.9 | high-middle | 21,029 | 4.67% | 184,635,668 | 9.17% |
| 0.7-0.8 | middle | 22,191 | 4.92% | 165,570,371 | 8.22% |
| 0.5-0.7 | low-middle | 45,490 | 10.09% | 280,444,336 | 13.92% |
| <0.5 | low | 348,969 | 77.42% | 1,206,252,321 | 59.89% |

Q59: Line 212: the slope toes on both sides of the river banks **are or get** erode**d**

A59: Thank you for your comment. We have corrected this expression in the sentence. Please see **Lines 217-218:**

Due to side bank erosion of rivers in river valleys, the slope toes on both sides of the river erode, leading to slope instability and landslides.

Q60: Line 213: for terrain reasons? Which means? Do you have any reference or measuring data to assess this?
A60: Thank you for your comment. "For terrain reason" actually means "considering the terrain condition". And we have revised the expression and added the reference to assess this. Please see **Lines 218-220:**
In terms of terrain conditions, the water vapor content in the air in valleys is high, and cold air masses flow into valley bottoms, forcing warm air to rise and resulting in rainfall (Li et al., 2021; Long et al., 2016; Kuwagata et al., 2001).
*Li G., Yu Z., Wang W., et al. Analysis of the spatial Distribution of precipitation and topography with GPM data in the Tibetan Plateau, Atmospheric Research, Volume 247, https://doi.org/10.1016/j.atmosres.2020.105259, 2021.*
*Long Q, Chen Q, Gui K, et al. A Case Study of a Heavy Rain over the Southeastern Tibetan Plateau. Atmosphere. 7(9):118. https://doi.org/10.3390/atmos7090118, 2016.*
*Kuwagata, T.; Numaguti, A.; Endo, N. Diurnal variation of water vapor over the central Tibetan Plateau during summer. J. Meteorol. Soc. Jpn. 79, 401–418. https://doi.org/10.2151/jmsj.79.401, 2001.*

Q61: Line 214: still avoid disaster – which implies more things happening that a "simple" landslide
A61: Thank you for your comment. We have changed carefully the following "disaster".

Q62: Line 232: have you done any statistical trend to assess the non-significancy?
A62: Thank you for your comment. We apologize for the unclear explanation. Please see **Lines 228-229**:
According to Fig.3, the number of grid units and geomorphic units with high landslide susceptibility in the eastern study area are more than the western area.

Q63: Line 235: amplification? Please change word
A63: Thank you for your comment. We have changed the word in the sentence. Please see **Lines 244-246:**
Figure 4 shows the **details** of the landslide susceptibility results the 10 quadrats A-L (excluding quadrats D and I) based on watershed and grid units. The details of the results for quadrants D and I are shown in the comparison of quadrat scales in the next section.

Q64: Line 234-245: It would be nice to see not only the difference between slope units and grid units, but also between slope units and real landslide and grid units and real landslides. Additionally, in figure 4, it would help to see the DEM below the slope and grid units to have an idea of the local topography.
A64: Thank you for your comment. We have analyzed the difference between geomorphic units and real landslide and the difference between grid units and real landslides. Please see **Lines 251-254:**

[Figure]

**Landslide susceptibility**

>0.9  0.8-0.9  0.7-0.8  0.5-0.7  <0.5

▼ actual landslide points

**Figure 4: Magnification of the contrast in quadrat A-L (except D and I) based on watershed and grid units.**

Q65: Table 2: please add lines to help the reader following rows.
A65: Thank you for your comment. We add lines in **Table 3**.

**Table 3.** Table of partition modelling precision

| region | observed | geomorphic unit | | | | grid unit | | | |
| | | predicted | | precision | accuracy | predicted | | precision | accuracy |
| | | Landslide | No Landslide | | | Landslide | No Landslide | | |
|---|---|---|---|---|---|---|---|---|---|
| A | Landslide | 181740 | 22516 | 89.0 | 84.4 | 164684 | 39572 | 80.6 | 78.4 |
| | No Landslide | 40793 | 161340 | 79.8 | | 48133 | 153997 | 76.2 | |
| B | Landslide | 125712 | 24084 | 83.9 | 82.7 | 123228 | 26568 | 82.3 | 78.0 |
| | No Landslide | 27370 | 121129 | 81.6 | | 39084 | 109415 | 73.7 | |
| C | Landslide | 74920 | 20840 | 78.2 | 74.5 | 75640 | 20120 | 79.0 | 74.2 |
| | No Landslide | 27095 | 65315 | 70.7 | | 28325 | 64085 | 69.3 | |

Q66: Lines 275-284: it is not clear, or has not been explained, how you used the google images to identify possible landslide points in areas with no actual landslides. So all of the discussion about figures 5 and 6 is no informative.
A66: Thank you for your comment. We have explained how the google images to identify possible landslide points. Please see **Lines 280-286:**
The dark red points in Fig. 5 and Fig. 6 are the landslide point, which are obtained from manually interpreted the Google Earth remote sensing images. According to the Remote sensing image, these landslide points are mostly the bare landform. There are dense vegetation around these points. The locations of these possible landslide points are within the centers of the potential landslide areas. Red points in Fig. 5 and Fig. 6 come from "The standard specification of geological hazard risk survey and assessment in China", which obtained from the fieldwork. These actual observation landslide points are concerned that may occur and are potentially dangerous to surrounding homes, roads, etc. The difference between dark red points and red points is that dark red points are less dangerous to houses and roads, so there are no dark red points in the database.

Q67: Lines 285-286: of course … they are based on the same units, why would there be overlapping?
A67: Thank you for your comment. This is a mistake in our presentation. We have revised this sentence. Please see **Lines 291-292:**
Based on the data presented in Figure 5(c), no landslide point is located within the low-landslide susceptibility geomorphic units.

Q68: Line 324: I am lost – please repeat here what unit scale refers to. Also, it would be nice to know at the beginning of the paper that you are going to do this other "test". Thus, not only compare the results of slope units and grid units but also explore the most important factors within your selected 16 influencing factors.
Dataset.

A68: Thank you for your comment. We have explained what unit refers to and make the readers know exploration of 16 influencing factors at the beginning of the paper. Please see **Lines 329-330** and **65-66:**

As shown in Fig. 7(a), one geomorphic unit from the unit quadrat was selected to analysis the detail distribution of influencing factors.
Moreover, this study analyzed the influencing landslide factors among two kinds of units.

Q69: The 3 shapefiles have different attribute tables. Names and number of fields do not correspond, and it is not clear what they refer to. As it is, the dataset is difficult to use by other people.

| Basin up | Basin mid | Basin down |
|---|---|---|
| id | gid | gid |
| gridcode | id | id |
| area | gridcode | gridcode |
| shape_leng | area | area |
| shape_area | shape_leng | shape_leng |
| huapo | shape_area | shape_area |
| nishiliu | huapo | huapo |
| bengta | nishiliu | nishiliu |
| fid1 | bengta | bengta |
| P | fidnew | fid_new |
| huapo1 | perimeter | Rowid_ |
| Rc | Rc | FID_1 |
| 市代码 | fid1 | FID_12 |
| area_level | pre_nsl | Rc |
| | | fid_12_13 |
| | | pre1_hp |
| | | fid_12__14 |
| | | pre_nsl |

A69: Thank you for your comment. We have modified the names and numbers of the attribute tables in the 3 shapefiles to keep consistent. And we have added the data description document to make others use conveniently.

| shapefile | Attribute table | | |
|---|---|---|---|
| regionA | area | P | landslide |
| regionB | area | landslide | P |
| regionC | area | landslide | P |

Area:the area of each geomorphic units. The unit is square kilometer.

Landslide: It indicates whether each geomorphic units has landslide points. In addition, "1" means yes and "0" means no.
P: It indicates the probability of landslide susceptibility assessment.

---

## Author Comment (AC2)

We would like to thank the referee for the time and effort they put in to review the first version of our manuscript. Their constructive comments enabled us to improve the quality and clarity of the manuscript. Actually, according to the suggestion of the RC1, we redefined our units into the geomorphic units which are more reasonable. We have checked the grammar and spelling errors of the manuscript. Please find our answers to the points raised below.

Q1: Line 15 (in the Abstract): „...topographic characteristics of different units comparatively faithfully, which is being increasingly extensively...” Here, you used too many, useless adjectives; you should rephrase this sentence.
A1: Thank you for your comments. We have rephrased this sentence. Please see **lines 11-14**:
In contrast to the grid units used in traditional spatial analyses, geomorphic units have explicit geomorphological and environmental implications and faithfully capture the topographic characteristics of different units; thus, geomorphic units have been applied in investigations of natural hazards, ecological processes and environmental impacts.

Q2: To enable more researchers to focus more conveniently on the subject matter to be addressed itself, rather than being caught up in the slope unit delineation”. This sentence does not have clear meaning for me, please rephrase.
A2: Thank you for your comments. We have rephrased this sentence. Please see **lines 16-18**:
This study enables more researchers to focus on hazard susceptibility assessments based on geomorphic units rather than the complicated process of geomorphic unit delineation.

Q3: Line 30: The sentence „Due to the intense tectonic activity and complex topography of the eastern edge of Tibetan Plateau, the deformation and failure of steep slopes are prone to slide” is not related to the previous one. It would be better to find another place and move this sentence, for example after the Line 55.
A3: Thank you for your comments. We have moved this sentence as your suggestion. Please see **lines 52-54**.

Q4: Line 82: What means „a relative height difference of more than 1000 m”? Is it the relative elevation between the top and the bottom of the canyon? Be more concise and use the specific terms for altitude values (either relative altitudes, or altitude a.s.l.).
A4: Thank you for your comments. We have rephrased this sentence to make it concise. Please see **lines 77-79**:
The crisscrossing and dense distributions of rivers have produced intense river erosion and deep canyon landforms, with relative elevations greater than 1000 m accounting for 81.5% of the total area (Li, 1989; Bian et al., 2018).

Q5: Lines 85-86: You describe here the climate conditions in the study area „with dry

winters, wet summers and obvious wet–dry seasons". This description is very imprecise and needs more details about the climate conditions at regional and local scales.

A5: Thank you for your comments. We have added details and rephrased it at at regional and local scales. Please see **lines 81-83**:

The eastern edge of the Tibetan Plateau is situated in a humid and temperate climate zone, and the average annual temperature is approximately 5°C. There is abundant rainfall, with annual precipitation greater than 500 mm. The precipitation in summer is 50% of whole year. The valley areas form a hot and dry climate affected by foehn wind.

Q6: the description of the landslide types (line 90) explaining that „They are mostly fast-moving slide-type and flow-type movements, and rapid-moving landslides are also abundant". This is, again, very imprecise and needs more clarification of the terms referring to landslides you used.

Multiple landslide types exist, depending on their type of movement. You should be more explicite and clearly mention in the description all the types of landslides occurring in your study area.

A6: Thank you for your comments. We have added the detailed description of all the types of landslides occurring in the study area. Please see **lines 86-89**:

The types of landslides in the study area are precipitation-induced landslides and earthquake-induced landslides. Rainfall-induced landslides are subdivided into the plane slip type, wedge outburst type, and structural collapse type. Earthquake-induced landslides are subdivided into the interlayer detachment type, brittle shear type, and brittle tension type (Liu, 2014).

*Liu C. Genetic Types of Landslide and Debris Flow Disasters in China, Geological Review, 60:4, 858-868, DOI:10.16509/j.georeview.2014.04.017, 2014.*

Q7: Lines 110-133: Here you mentionned the 16 conditionning factors taken into consideration to assess the landslide susceptibility. But, how did you chose to analyze the type and number of these conditionning factors? For different lanslide types you should consider to analyse different conditionning factors. A conditionning factor favoring a type of landslide will not necessary favor other landslide type. Knowing the different landslide types occurring in the study area is therefore essential, to select the conditionning factors for each landslide type that should be included into the susceptibility analysis.

A7: Thank you for your comments. For different landslide types we should consider to analyze different conditionning factors. At the beginning we selected 30 influencing landslide factors, according to the existing study and landslide type in the study area. Finally, the results of 16 influencing factors are significant. Please see **lines 111-114**:

At the beginning we selected 30 influencing landslide factors, according to the existing study and landslide type in the study area. The results of 16 influencing factors are significant. It shows as the reference in Table 1.

Q8: Lines 270 and 285: Here you mentioned the „actual observed landslide points". I understand that the landslides in your analysis are located as points on your map. How did you chose the location of the point, is it within the centre of the the landslide area? All grid units statistics are conditionned by the way in which you located the landslide point on the map. A particular grid unit will contain, or not a landslide point, depending on how this point was located on the map. You should provide detailed information about the location of landslide points and how the location of these point

A8: Thank you for your comments. We have added the information about the location of landslide points and how the location of these points. Please see **lines 277--279**:

In addition, the location of the actual observed landslide points is the trailing edge of the landslide according to the "The standard specification of geological hazard risk survey and assessment in China". The actual landslide points were obtained from the fieldwork project of the China Geological Survey.